# A Monolithic Gimbal Micro-Mirror Fabricated and Remotely Tuned with a Femtosecond Laser

**DOI:** 10.3390/mi10090611

**Published:** 2019-09-14

**Authors:** Saood Ibni Nazir, Yves Bellouard

**Affiliations:** Galatea Lab, STI/IMT, Ecole polytechnique fédérale de Lausanne (EPFL), Rue de la Maladière, 71b, Neuchâtel CH-2002, Switzerland; yves.bellouard@epfl.ch

**Keywords:** femtosecond laser machining, non-contact, tunable optics, micro-optics, flexures, repositioning, monolithic, fused silica, integrated optics

## Abstract

With the advent of ultrafast lasers, new manufacturing techniques have come into existence. In micromachining, the use of femtosecond lasers not only offers the possibility for three-dimensional monolithic fabrication inside a single optically transparent material, but also a means for remotely, and arbitrarily, deforming substrates with nanometer resolution. Exploiting this principle and combining it with flexure design, we demonstrate a monolithic micro-mirror entirely made with a femtosecond laser and whose orientation is tuned in a non-contact manner by exposing some part of the device to low energy femtosecond pulses. Given the non-contact nature of the process, the alignment can be very precisely controlled with a resolution that is many orders of magnitude better than conventional techniques based on mechanical positioners.

## 1. Introduction

Most optical devices are sensitive to precise alignment between their various elements. In some cases, even a small degree of misalignment—albeit small and comparable to a fraction of the wavelength, can lead to a degradation in performance or even operational failure. Regardless of the fabrication tolerances, inherently present in any manufactured product, most optical circuits are prone to misalignment due to environmental factors, such as temperature fluctuations or mechanical vibrations. While such environmental constraints can be mitigated in a laboratory environment, in a fully assembled product, it is imperative to achieve a very high degree of stability against such factors in field operation.

Here, we investigate novel concepts for integrating free-space optical components in a way such that high accuracy permanent alignment is achieved, in a contact-less manner, without the need for adjusters. In this particular work, we focus our attention on micro-mirrors, one of the most basic components in an adaptive optical system.

In industrial environments, alignment errors can creep in during the assembly process itself. Such errors can have multiple origins. One of them is a lack of accuracy in the positioning tool used to place an element with respect to the other. A second one is related to fabrication tolerances of each of the components to be assembled, and, finally, a third one may originate from the attachment method used. For instance, techniques such as, soldering, gluing or brazing—common in assembly processes—create interfaces between materials with different properties, such as different thermal conductivities, coefficients of thermal expansion (CTE) and Young’s modulus [1]. Consequently, this leads to post-assembly shifts or built-in stresses. In the case of an optical system, these issues can be dramatic, due to usually high alignment accuracy requirements, typically approaching a fraction of the optical wavelength. For all these reasons, translating a laboratory optical system into a reliable assembled product can be a daunting task, requiring dedicated assembly methods and designs.

Various ideas have been proposed to solve these issues, including post-assembly corrective actions and pre-compensating whenever possible during assembly [2,3].

Among these methods, localized laser thermal melting offers a non-contact alignment approach [4,5], but suffers from intrinsic limitations, such as the presence of heat affected zones and design-limitations, as it relies on a *linear* absorption process and, consequently, can only be applied to the surface of materials.

To cope with these issues, laser-shock adjustment with short pulses has been proposed [6,7,8]. While still relying on linear absorption events, the use of short pulses lowers the timescale of interaction and minimizes thermal diffusion effects. However, it still remains a surface interaction effect and induces localized stress concentration.

One step further is to use ultrashort pulsed laser-matter interaction that allows for *non-linear* absorption events to occur with *no heat transfer* to the surrounding matter—at least, not beyond the area under *direct* laser exposure. Applied to transparent materials, thanks to the rich-taxonomy of complex material transformations occurring under ultrafast laser exposure [9], this effect can be used inside the bulk to create localized volume expansion or shrinkage of micron-size volumes (‘voxels’) and, thus, resulting in displacements in the nanometers range [10,11,12].

In addition to localized volume modifications, femtosecond laser exposure together with wet chemical etching offers a very versatile three-dimensional manufacturing process [13,14]. Starting with a single substrate, structures of any given shape can be created almost anywhere inside the specimen volume. Particularly in fused silica, the process has been used to create both optical as well as pure mechanical devices. Active and passive waveguides [15,16,17,18], couplers [19,20], diffractive optical elements [21,22], wave plates [23], and polarizing optics [24,25] have been demonstrated, as well as mechanical devices like actuators [26,27] and flexures [28]. The integration of optical and mechanical functionalities has also been demonstrated inside a single substrate, while maintaining the monolithic nature of the overall device [29]. Cheng et al. have demonstrated a micro-optical circuit machined inside photo-sensitive glass [30]. It is therefore possible to realize an optical circuit in a single chip of glass, where passive components are fabricated in place with a femtosecond laser.

Monolithic fabrication process, to a large degree, removes the need of post-fabrication alignment as by definition there is no assembly. However, as inherent to any fabrication method, manufacturing tolerances still remain, and are critical when aiming at sub-100 nm accuracy in a permanent and stable manner. Post-fabrication alignment thus remains a necessity.

The possibility of making tunable integrated optical devices is still an ongoing endeavor. Not only does it push further the idea of device miniaturization, it also offers more flexibility, higher precision and better stability over time and in harsh environments.

In the present work, we show that in-volume modifications resulting from femtosecond laser exposure, combined with monolithic flexures in a single fused silica substrate, can lead to unprecedented levels of angular positioning accuracy, and this, in a permanent manner. Building up on our previous work [31], here we specifically demonstrate a gimbal mirror with its own flexure positioning element, entirely made of fused silica, and whose orientation is locally adjusted by a laser and in both directions.

## 2. The Device

### 2.1. Concept and Working Principle

The proof-of-concept device, shown in Figure 1A, consists of a mirror surface (gold-coated to make it reflective) and a flexural element, all made-out of a single silica substrate. The mirror surface is suspended on two thin intersecting beams, and connected to a rigid bar (‘the actuator’) through a lever mechanism. The actuator is a wide and long cantilever, fixed on one end, which contains laser-modified regions as explained later. On the other end, it is connected to the amplification mechanism through a thin bar. To align the mirror in plane, the ‘actuator’ can rotate the mirror in either direction. The method of actuation is non-contact and permanent. It is based on femtosecond laser-induced volume changes [10] inside the bulk and provides sub-nanometer linear displacement resolution. The detailed functioning for the actuation principle is explained in another paragraph. Combined with specific flexure kinematics (Figure 1B), the linear displacement of the actuator is converted into a pure rotational motion of the mirror surface.

As a substrate material, fused silica offers unique advantages for a chip-based monolithic optical assembly. Apart from having one of the lowest known coefficients of thermal expansion, it is optically transparent over a broad range of wavelengths and is chemically inert, making it suitable for a variety of applications.

### 2.2. Kinematics

The kinematics of the gimbal mirror are shown in Figure 1B. Each pivot joint represents one degree of freedom. The center of the mirror—denoted here as R, is defined by the intersection of the two thin beams oriented at 45 degrees with respect to the mirror normal. The mirror is further supported by a thick arm (e) at its center that is used to transfer the force from the actuator. The latter, shown in red here, is exposed throughout its volume to short pulses to induce a net strain. As one end—the anchoring point of the actuator—is infinitely stiff in comparison to the other end, these volume changes are directed towards the less stiff end, thus causing a net displacement. A classic lever mechanism is employed to amplify this motion. An amplification ratio of 15 is chosen here to get an appreciable rotation of the mirror surface. Another thin beam couples this linear displacement to the mirror, effectively pulling or pushing on the beam and causing the mirror to rotate about its center. The mechanism connecting the actuator and the rotating mirror flexure is similar to a slider-crank mechanism. The two beams (a, c) are designed to be loaded in both tension or compression, without any appreciable effect on the overall functioning of the mechanism. For illustrative purposes, a deformed state of the device, corresponding to the situation where shrinkage occurs inside the actuator, is shown in dashed lines underneath.

### 2.3. Analytical Model

In this section, we present a brief analytical model of the flexure mechanism, whereby we define our system mathematically to better understand its response and to provide further optimization tools. Such an analysis is helpful to understand how the system will respond under the influence of an external force or moment. At the same time, it also helps to characterize the effect of various flexure dimensions and immediately identify the most critical parameters for ease of design.

In our analysis, we use six-by-six stiffness matrices to estimate the elastic deflection of the mechanism under a given loading condition. The primary governing equation is given by:
(1)$${\left(F^{p}\right)}_{q}={\left(K^{p}\right)}_{q}\times {\left({\hat{r}}^{p}\right)}_{q}$$
where *(F^p^)_q_* represents a wrench (a six-coordinate vector containing force and moment), *(K^p^)_q_* represents the stiffness matrix and *(r^p^)_q_* denotes the overall displacement of the system at any arbitrary point p, all represented in the same coordinate frame *q*. These equations can become very complex to handle, however, one can simplify the calculations manifold by representing the individual stiffness matrices in the principal frame of reference, and then use transport equations to calculate them at any other location. In such a representation, all the off-diagonal elements are zero. Once the overall stiffness matrix is known, for any given force, the net displacement can be calculated using Equation (1).

For a thin cantilever beam with length *l*, width *w* and thickness *d*, the principal stiffness matrix is given by the von Mises stiffness matrix [32]. The von Mises matrix can be further adapted to the case of a notch hinge too, although the individual terms are more complex. However, under the approximation *β* = (*d*/2*r*) << 1, the terms can be simplified further as is discussed in detail in the annex. Here, *d* and *r* denote the thickness at the center and radius of the hinge, respectively.

The global stiffness matrix for the entire system is obtained by combining unit stiffness matrices, such as the von Mises one. To do so, we transport all the stiffness matrices to point O (Figure 1A) and rewrite them in the global coordinate frame, the details of which are presented in the annex (refer to Figure A1). In particular, the overall translational stiffness along the x direction writes as:(2)$$K_{xx}=E\left(\frac{1}{l_{1}}+\frac{2}{l_{2}}+\frac{d^{2}}{l_{1}^{3}}+\frac{1}{\pi \sqrt{rd}}\right)dw$$

Here, *l*_1_, *l*_2_ represent the lengths of the inclined beams and the straight beams, respectively, *w*, *d* denote their width and thickness and *E* is the Young’s modulus of fused silica. Since *d* is much smaller than *l*_1_ and with the assumption that *l*_1_^²^ ⁓ *π^*²*^rd*, this expression can be further rewritten as: (3)$$\frac{K_{xx}}{E}=\frac{2dw}{l_{1}}\left(1+\kappa \right)$$
where *κ* denotes the ratio of *l*_1_ and *l*_2_. Using this equation, we can immediately see that in order to reduce the overall stiffness acting against the volume changes, the ratio *w*/*l*_1_ should be made as small as possible without compromising other stiffnesses.

### 2.4. Finite Element Modelling

We used a commercial FEM simulation software (Solidworks Premium 2017 ×64 Edition from Dassault Systems, Vélizy-Villacoublay Cedex, France) to optimize the design and flexure kinematics. First, a design analysis was performed to optimize the different parameters of the device. The dimensions of the flexures were chosen to minimize the overall stiffness in the direction of interest (according to Equation (3)), while maximizing it along other directions, thus limiting parasitic movements. Further, a static analysis was done as illustrated in Figure 2. For this simulation, a conservative 500 nm displacement was applied to the actuator along its length. The dimensions of the flexures were then iterated to maximize the rotation of the mirror for the applied displacement. As predicted by the simulations, the maximum stress in the device is of the order of a few MPa. We have shown previously that flexures manufactured with femtosecond laser processing can withstand a tensile load in excess of 2 GPa [33]. Note that here we consider a conservative value for the maximum stress. If we allow for higher stress levels, the design can be made more compact (i.e., occupying a smaller footprint) and significantly stiffer if one aims for higher resonance frequencies.

A dynamic study was also conducted to determine the natural vibrating modes of the device. As expected, there is not a very strong decoupling between in-plane and out-of-plane modes. This is due to the small thickness of the sample and the long length of the beams supporting the mirror. The vibrating modes are shown in Figure 2c. The primary out-of-plane mode occurs at the lowest frequency of 1.7 kHz, while in-plane modes occur at slightly higher frequencies.

Based on the mathematical analysis and the results of the simulations, the following parameters were chosen for the dimensions of the various flexures as listed in Table 1. However, the thickness (*w*) of the device was fixed by the available substrate to be 1 mm.

Substituting these values into Equation (2), we can calculate the overall translational stiffness along the *x* direction, and we find K_*xx*_ = 7 × 10^6^ N/m, which is in close agreement to the simulated value of 7.15 × 10^6^ N/m.

### 2.5. Fabrication Process

The device is fabricated by drawing the contour in a 25 mm × 25 mm × 1 mm silica substrate (synthetic fused silica with high OH content), which is mounted under the focus of an amplified femtosecond laser system (Yb-fiber, Amplitude Systèmes, Pessac, France), delivering 270 fs pulses at 1030 nm. The laser is focused by a 20× objective (N.A 0.4) mounted on a vertical stage. The sample is translated under the focus of the laser with the help of a precise positioning stage (Micos, Ultra-HR, Eschbach, Germany). Starting from the lower surface of the substrate, laser affected zones are stitched together as planes, gradually moving up until the top surface is reached.

After the exposure, the edge containing the mirror is polished to optical quality and the sample is placed in a bath of dilute Hydrofluoric acid (2.5%) for ⁓30 hours, thus dissolving the laser-affected zones (LAZs). As a final step, the edge is gold coated to make it reflective. For the fabrication, a pulse energy of 250 nJ and a writing speed of 8 mm/s is used with the repetition rate set at 750 kHz. The device is shown in Figure 3.

## 3. Results

### 3.1. Exposure Setup

The magnitude of the strain induced by femtosecond pulses is a function of various parameters such as pulse energy, spot size and deposited energy (net fluence). Depending on the pulse width, volume expansion (positive strain or ‘regime II’ as referred in the literature [34,35]) is observed for long pulses while as an overall reduction in volume (negative strain or ‘regime I’ [15]) occurs for short pulses with low pulse energy. However, in terms of magnitude, it ranges from 0.01% to 0.05% of the exposed volume. Under our experimental conditions, this corresponds to a few hundred nanometers of linear motion of the actuator.

To measure the angular rotation of the mirror, we use a triangulation scheme as shown in Figure 4. A fiber pigtailed laser diode at 980 nm is used as a light source. The laser beam is focused and then re-collimated such that the spot size is less than 1 mm on the test-mirror surface. This ensures that the beam is contained completely within the mirror and does not diffract off the edges. The reflected beam then passes through an ‘*f*-*θ* lens’ and is focused on a position sensing device (PSD, SiTek Electro Optics, Partille, Sweden), that measures the actual position of the beam in a plane, therefore, not only measuring lateral, but also vertical motion. The *f*-*θ* lens, in addition to amplifying the output displacement, has a flat image plane at the position of the detector, thus ensuring the beam is always focused in the same plane. By design, the displacement in the image plane is *linearly* proportional to the change in angle of incidence, and is given by:(4)Δx=f×Δθ

Here, ∆*θ* is the change in the in-plane angle of incidence of the incident ray with respect to the optical axis of the lens, *f* is the focal length of the lens and ∆*x* is the net displacement observed on the surface of the detector.

Similarly, any change in the out-of-plane angle can also be measured by recording the movement of the focal spot along the y-axis. The two are then related as:(5)Δy=f×Δϕ
where ∆*∅* is the change in the out-of-plane angle of incidence with respect to the optical axis of the lens.

The entire assembly is then mounted on a precision *x*-*y* stage and under the focus of a short pulse laser. The actuator is exposed by writing a series of closely spaced lines in the bulk. Starting near the bottom surface, a 1 mm-wide plane of laser-affected region is written. This plane consists of 2.5 mm-long parallel lines separated horizontally from each other by 2 μm. The laser focus is then moved up 11 μm, gradually stacking the lines on top of each other. To prevent surface ablation, we leave a buffer of 50 μm, both on top as well as on the bottom surface, effectively reducing the height of the exposed region to 900 μm. Due to optical aberration effects, the volume change is not uniform throughout the bulk of the exposed region [36]; the absorption decreases as one goes deeper into the material and expectedly so, thus causing non-uniform strain. This can be simply compensated for by either ramping up the power with depth or by writing fewer lines near the top surface, thus creating a uniform modification throughout the bulk. We follow the latter strategy and write the laser-affected zone as a trapezoidal block, gradually decreasing the number of lines as we move towards the top surface by 20%. Another alternative—albeit more costly, is to use adaptive optics to correct for spherical aberrations [37].

### 3.2. Identification of Optimal Writing Parameters

Here, we explore exposure conditions that will lead to either a shrinkage or expansion of the actuator, so that a bidirectional motion can be obtained. Indeed, it has been shown that an interplay between the pulse width and/or the pulse energy can reverse the sign of the overall strain [38]. In our case, it presents an opportunity to achieve bi-directional motion of the mirror.

To identify the optimal exposure parameters, it is important to know the relationship between the overall strain and the deposited energy for different pulse widths. Such a relationship is necessary to predetermine the amount of strain that will be generated, thus avoiding cracks/failure due to excessive stress or limited to no deflection due to a too low a stress level. To do so, we follow a method developed earlier by our group [10]. A series of cantilevers are fabricated on a 250 μm-thick sample of fused silica. These cantilevers are 18 mm-long and 1 mm-wide. After chemical etching, they are loaded near the anchor point by writing a series of parallel lines 35 μm below the surface. The exposure is carried out for two pulse widths—50 fs and 300 fs—and only near the top surface, thus forming a bilayer of modified and unmodified material. For each fixed pulse energy, the deposited energy is gradually increased by reducing the writing speed, as shown in Figure 5. Depending on the pulse width, volume expansion is observed for long pulses while as an overall reduction in volume occurs for short pulses. This change in volume is amplified by the length of the cantilever causing the other end to bend out of plane. Afterwards, the deflection of the cantilever tip is measured using a white light interferometer and the strain is calculated using:(6)$$\varepsilon_{laz}=\left[\frac{\delta }{L_{l}\left(L_{c}-L_{l}/2\right)}\right]\left(\frac{h_{laz}l_{s}}{w_{{}^{\circ}}}\right)$$

Here, δ is the total measured deflection, *h_laz_* is the height of the exposed region from the neutral line, *l_s_* is the line spacing, *L*_l_ is the length of the exposed region, *L_c_* is the length of the cantilever, and *w˳* denotes the width of a single modification. The details of this method can be found in [10].

Based on this data, we can predict the overall deflection of the mirror and, more importantly, stay within the elastic limits. For shorter pulses, we see a monotonic growth in the strain as the deposited energy increases towards higher pulse energies. However, for longer pulses, we confirm an already established trend, where the strain peaks around a fluence of 10–20 J/mm^2^ before declining sharply [39]. It is also important to note that at higher pulse energies, the strain observed at longer pulse lengths is much larger as compared to that of shorter pulses. Later on, we see that this translates directly into a larger angular rotation of the mirror.

### 3.3. Measurement with Short Pulses (50 fs) 

For this experiment, we use an optical parametric amplifier (OPA) (from Amplitude Systèmes), which emits short pulses with a pulse width of 550 fs which are then compressed to 50 fs by passing them through a prism compressor. The seed laser for the OPA is an ytterbium-based femtosecond source, emitting 300 fs laser pulses at a 120 kHz repetition rate with a maximum average output power of 2.6 W. 

The first exposure is carried out close to the actuator fixed end as shown in Figure 1. For this exposure, a pulse energy of 260 nJ and a writing speed of 1.38 mm/s is used, thus having a net fluence of 20 J/mm^2^. To ensure a uniform deposition of energy across the actuator width, a buffer of 500 μm is kept on both sides. This allows enough time for the x-y position stages to accelerate to a constant speed before the beam is effectively scanned across the material. As the exposure process starts, the position of the probe laser beam reflecting off the mirror is continuously monitored on the position sensing device. The readouts from this device are the x and y coordinates of the laser beam, which are measured within an accuracy of ±1 μm on the detector surface. The final width of the laser affected zone is 1 mm at the bottom and 0.68 mm at the top, and consists of ~34,000 lines arranged in a trapezoidal shape. As the material gets exposed, the actuator begins to shrink in volume, thus rotating the mirror anti-clockwise. In this experiment, the exposure process takes approximately 27 hours. Note that this time could be considerably reduced by shaping the beam and increasing the writing speed, for instance by scanning. The results are presented in Figure 6.

Along with the in-plane rotation, an out-of-plane parasitic movement is also observed (marked as Y, *θ*_v_ in Figure 6a. This effect is attributed to the writing strategy, which follows a sequential pattern of bottom-to-top writing of successive planes. As the laser starts to write near the bottom surface, the exposed volume contracts (or expands depending on the pulse duration), while the top surface remains unaffected. This creates a bimorph-like layered structure of modified and unmodified material, causing the actuating element to bend down during this sequence. Next, as the laser focus moves up and crosses the neutral line, the strain in the lower layers is gradually compensated, and the bending effect is cancelled, leaving only a net contraction (or, conversely for longer pulses, an expansion) of the actuating element without out-of-place deflection.

The angular position of the mirror is back calculated from the measured change in the x and y coordinate of the spot using Equation (6):(7)$$\Delta x\;=\;2\times f\times \Delta \theta_{h,mirror}\;\mathrm{and}\;\Delta y\;=\;2\times f\times \Delta \theta_{v,mirror}$$
where *f* is the focal length of the focussing lens, and the indices *h*, *v* denote the horizontal and vertical rotations, respectively.

As any out-of-plane motion is amplified manifold by the length of the actuator, resulting in a magnified movement at the other end. Consequently, to reduce the amplitude of the parasitic movement appearing during the imprinting of the pattern, we expose the actuator again, this time away from the anchor point and close to its far end. As expected, the parasitic movement is vastly reduced as shown in Figure 7.

It can also be seen that by moving the LAZ to the far end of the actuator, the angular rotation is almost doubled, for the exact same exposure conditions. This effect can be attributed to the fact the nearly all of the strain is transferred to the mobile mirror, since the stiffness of the unexposed region of the actuator itself no longer comes into play. In other words, less of the strain energy is lost in the actuator bar itself.

### 3.4. Measurement with Long Pulses (300 fs)

We repeated the experiment in a new sample, however, with a longer pulse duration to access the modifications in regime II, i.e. the so-called nanogratings.

For this part of the experiment, we bypass the prism compressor and the OPA to directly use the output of the seed-amplifier. For the exposure, we use a pulse energy of 160 nJ and a deposited energy of 8 J/mm^2^. The whole process is then repeated as earlier. The results are presented in Figure 8.

As observed with the cantilever experiments (Figure 5), the ratio between the measured values of strain for the deposited energies used is about 1:3 between short and long pulses. A similar ratio is obtained in the total deflection angle measurement between the two pulse widths.

As can be seen from Figure 8a, and as expected, the mirror movement is reversed as compared to earlier experiments with *τ* = 50 fs, thus confirming an overall volume expansion (positive strain) inside the actuator. In addition, we see a vertical offset of nearly 53 μ-radians between the initial and final out-of-plane angle (Figure 8b). This is due to a large in-plane deflection angle causing the beam to travel beyond the active area of the detector (marked by a cross). As discussed in the previous section, any parasitic offset can be corrected for by modifying the writing strategy and writing more lines near the top surface, as in this case, or writing fewer, as in the earlier cases. In general, out-of-plane bending can be nearly completely suppressed by writing close to the neutral line of the actuator rather than exposing the whole bulk from bottom to top.

Further, we observe a slightly non-linear behavior in the variation of position with time (Figure 8a). We attribute this effect to an increased loading along the axial direction of the pivoting hinge (in the crank-wheel kinematics part of the mechanism) as the magnitude of the strain starts to be significant. Indeed, it is known that a hinge sees its own bending stiffness reduced under the effect of transverse load [40]. In other words, a non-linear coupling takes place. As the actuator expands, an increasing load is applied to the pivots (referred to as d and, to a lesser extent, a, in Figure 1) that see their own transverse stiffness lowered and as a result deform more for the same amount of strain applied to them, eventually causing the mirror to deform more (hence, the increase of slope in Figure 8a).

### 3.5. Stability

In this section, we explore the stability of the device once the actuator is loaded. The idea is to test whether the mirror can hold its position in a deformed state without relaxing back to its normal position or to see if there is any material relaxation effect to be expected. Any such relaxation, if observed, could be caused by multiple factors, such as temperature variations, other environmental factors, or the decay in stress itself—either in the flexures or in the laser affected zones. Since fused silica has one of the lowest known CTEs, it is highly unlikely that temperature variations (in our lab, it is stable within ±2°) could play a role.

To understand this, we load our device with the same parameters as used in last experiments (Section 3.4). After exposure, the deflection is measured under a white light interferometer as shown in Figure 9c. For the stability test, we use a digital holographic microscope (Lyncée Tec, Lausanne Switzerland) in a reflection mode that illuminates both the mirror and the static glass frame at the same time while holograms are recorded. Based on the recorded pattern, the change in height difference between any arbitrary point on the mirror surface and a fixed point on the base is monitored for 16 days. In Figure 9a, the data for four such points is plotted.

As can be seen, we do not observe any trend that would suggest any departure from the deformed state, i.e., the mirror holds its place. Though the test is very specific to this device, it does not aim to understand the general phenomenon of relaxation or failure that occurs in glasses subjected to high stresses. Although predominant at high stress levels (close to 1 GPa or more), there seems to be almost no decay at low stress levels [41], with the peak stress not exceeding a few tens of MPa in our case. It is important to mention that this decay also depends on environmental conditions such as the surrounding humidity.

## 4. Discussion

In this paper, we have investigated the repositioning of a laser beam in a non-contact manner using femtosecond laser-assisted structural modifications in fused silica. We have shown that by changing the pulse width along with a proper selection of pulse energy and deposited energy, a reversible motion can be achieved. Although the case is demonstrated in two different samples, the same can be achieved inside a single specimen, as well.

We have further shown that such repositioning, in addition to being non-contact, is also permanent at the present stress levels. This method of non-contact alignment is most suited for very fine repositioning and, thus, does not risk very high stress levels. At very high stresses (a few hundred MPa or more), relaxation is a real problem, with no concrete law explaining how the decay occurs.

To further optimize this method, it is necessary to reduce the overall time it takes to do the realignment. With high pulse energies, high repetition rate lasers and by using fast scanning mirrors, the overall time could be reduced to a fraction of the current time. Further, the long pulse laser already reduces the repositioning time by more than 75%. In most high precision optical devices, a high degree of alignment is achieved in the fabrication process itself, so what needs to be corrected for is much smaller than what is demonstrated here. In the end, we estimate that the alignment time could be reduced to just a few minutes, much like a conventional mirror mount, although permanently and with a much higher precision.

Although, a single degree of freedom is explored here, with different kinematics, it is possible to achieve a controlled movement along other directions too, without changing the underlying method.

Finally, femtosecond lasers have been widely used for the fabrication of various structures, both optical and non-optical. The ability to fine tune such elements, in a non-contact manner, without the requirement of any sophisticated tools and with unprecedented resolution is a step towards the larger idea of fully monolithic optical circuits.

## Figures and Tables

**Figure 1 micromachines-10-00611-f001:**
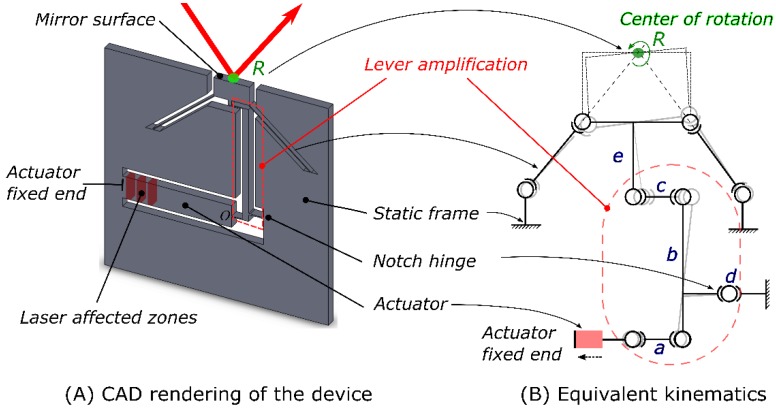
(**A**) Computer-aided design (CAD) drawing of a proof-of-concept of laser-tunable gimbal mirror. (**B**) Schematic of the mirror kinematics. The strain inside the actuator is coupled through a thin beam (a) to the lever arm (b). A second thin beam at the top (c) ultimately pulls on the mirror through (e) to cause a rotation. The lever mechanism is implemented through a circular notch hinge (d) fabricated by the same process. As the actuator is exposed, there is a net displacement towards the left and a subsequent rotation of the mirror. A deformed state of the mirror is shown in light gray.

**Figure 2 micromachines-10-00611-f002:**
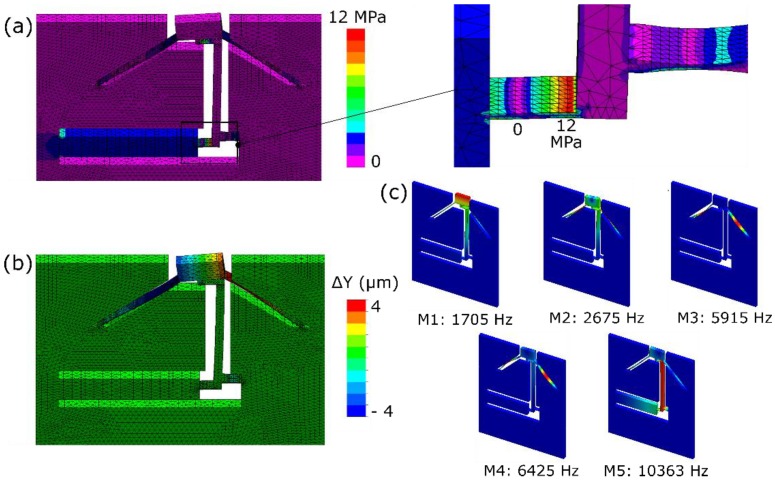
Results of the prescribed displacement analysis simulation corresponding to a 0.5 μm displacement applied to the actuator showing the (**a**) corresponding stress plot with a maximum stress level equal to 12 MPa and (**b**) total displacement along the *y*-axis (the figures are in the *x*-*y* plane with the *z* axis pointing outwards and *y* axis pointing upwards). (**c**) Natural vibrating modes of the device (Note that the vibration-induced displacements are exaggerated for visualization purposes).

**Figure 3 micromachines-10-00611-f003:**
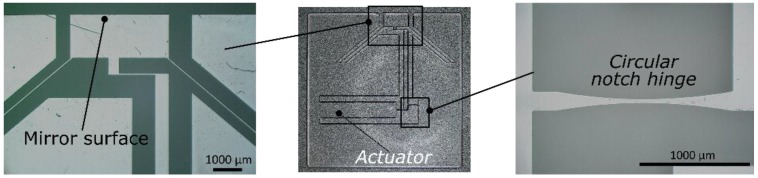
The fabricated device is shown here (**middle**). A digital microscope image of the mirror edge (**left**) and the notch hinge (**right**) are also shown.

**Figure 4 micromachines-10-00611-f004:**
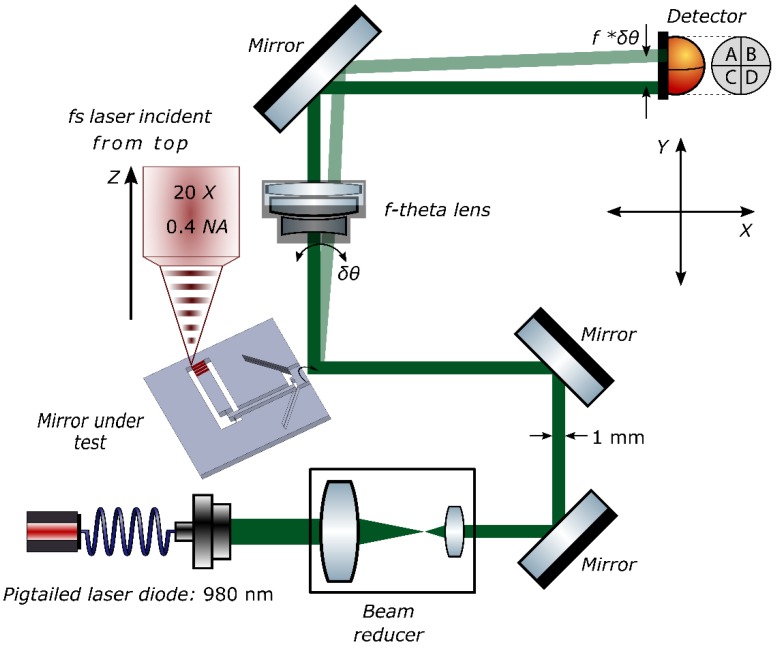
A schematic of the experimental test bench. A probe laser beam is focused, re-collimated and reflected off the test mirror. The reflected beam is then focused by an f-theta lens on a position sensing detector (PSD) that records its position in real-time. The whole assembly is then moved under the focus of a femtosecond laser and a sequential pattern of laser affected zones is written inside the actuator. The modifications induce a net strain causing the mirror to rotate about its center. The subsequent change in the position of the focal spot is then recorded and monitored on the detector.

**Figure 5 micromachines-10-00611-f005:**
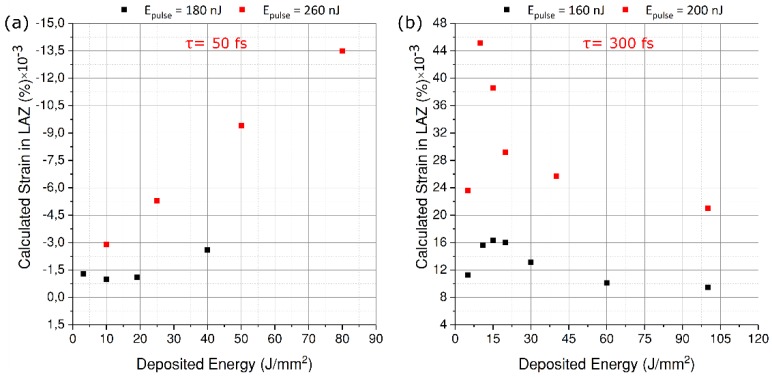
Strain in the laser-affected zones as a function of deposited energy for (**a**) τ = 50 fs and (**b**) τ = 300 fs and for two different pulse energies. One should note the switch from regime I (**a**) to regime II (**b**) with the increase of the pulse duration.

**Figure 6 micromachines-10-00611-f006:**
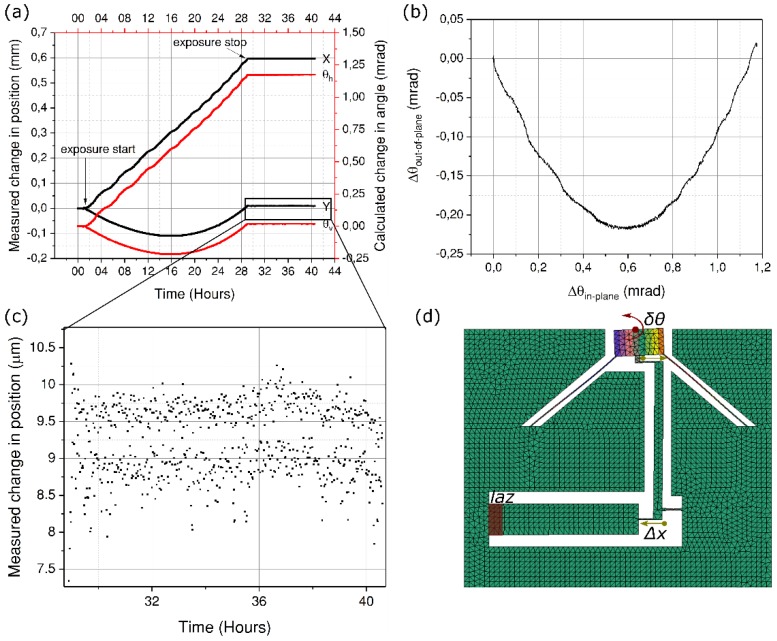
(**a**) Position measurement of the laser probe-beam spot on the detector surface. The measured change in position is plotted in black, while the corresponding angle change is plotted in red. (**b**) Out-of-plane bending angle vs the in-plane angle of rotation. (**c**) A magnified view of one of the position coordinates (y) after the exposure is stopped to show case the stability of the mirror in its deformed state. The observed noise is within the error of the detector. (**d**) The exposure configuration.

**Figure 7 micromachines-10-00611-f007:**
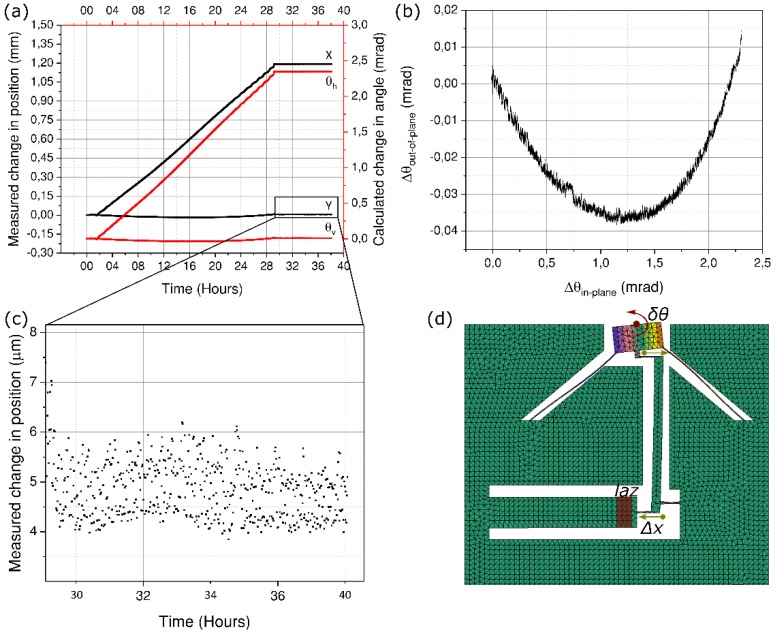
(**a**) Experimental observation of the spot position when the actuator is exposed at the far end away from the anchoring point. Plots (**b**–**d**) follow their respective plots in Figure 6.

**Figure 8 micromachines-10-00611-f008:**
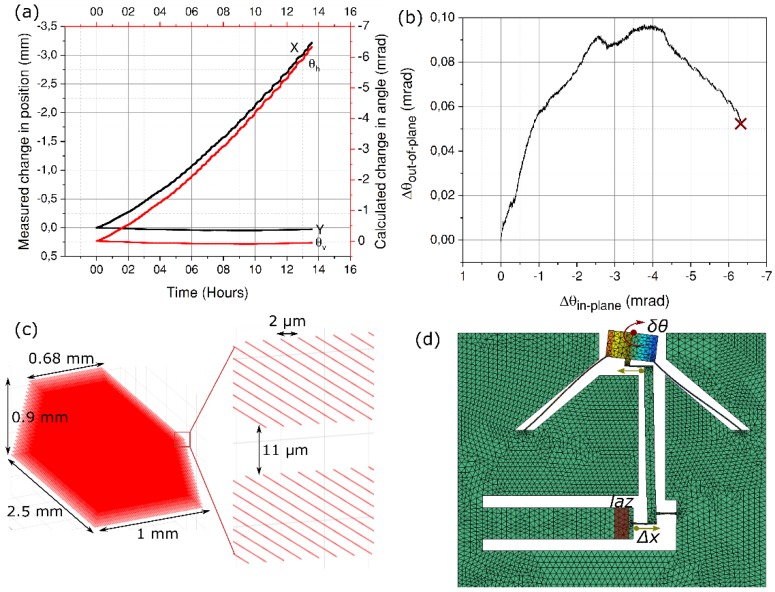
(**a**) Experimental observation of the laser beam spot on the detector surface when the actuator is exposed to long pulses at *τ* = 300 fs. (**b**) Out-of-plane bending angle vs. the in-plane angle of rotation. (**c**) A 3D representation of a single laser affected zone used in all the experiments. In the inset, two vertically separated planes containing the lines are shown. (**d**) The exposure configuration. The direction of strain is reversed here as is pointed by the arrows.

**Figure 9 micromachines-10-00611-f009:**
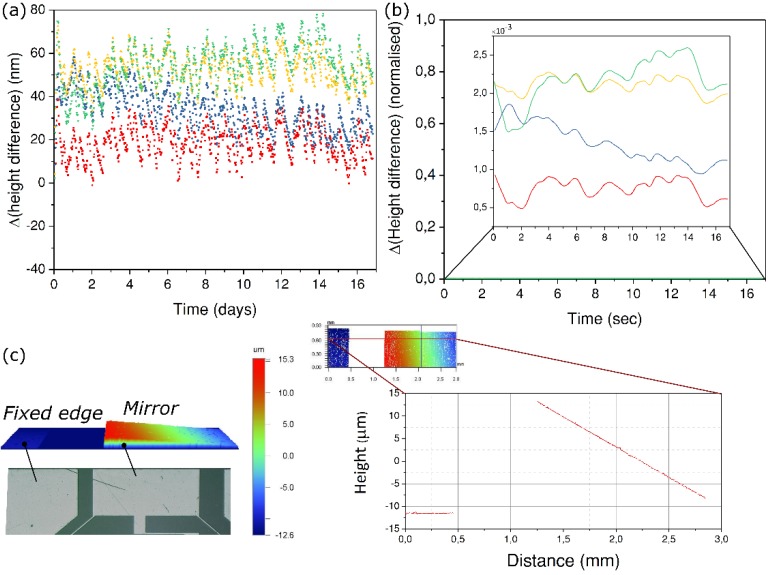
Stability analysis of the mirror in the deformed state. (**a**) The change in height difference between four arbitrary points on the mirror surface and a fixed point on the static edge. (**b**) The normalized variation. (**c**) A 3D color map of the deformed mirror (left) and a line plot of the height variation across the mirror (right). The height difference between the mirror and the base is used to normalize the data in (**b**).

**Table 1 micromachines-10-00611-t001:** Main design parameters.

*w* (µm)	*d* (µm)	*l*_1_ (mm)	*l*_2_(mm)	*r* (mm)
1000	50	9	1.2	4.5

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
