# Peer review of "A Monolithic Gimbal Micro-Mirror Fabricated and Remotely Tuned with a Femtosecond Laser"

_micromachines, 2019, doi:10.3390/mi10090611_

Round 1
Reviewer 1 Report
1 can be separated as (a) and (b). 2 (c) is overlapped, and there is no color code for the . Also the stress distribution (a) does not show where the max static stress 12MPa is. 1 Exposure Setup seems to be in section 2. Line 209 repeated words. The whole paragraph should be rewritten for clarity. The function of the f-theta lens is not clear. It may change the deflected angle to be measured. The actuator is not explained in the manuscript. I wonder if the femto second laser is used for fabrication purpose or as a driving force. It’s confusing. If for driving purpose, using ultrafast laser is an over-kill. I can only review up to this stage without the above information is cleared for me. The author did lots of work and the methods seem to be sound overall, despite some contents can be more properly presented. The most important is that the goal of the whole research is not clear. The fs laser is used for fabrication without any doubt, however, more than half of the experiments are about "driving" the mirror. The authors should make it manifest to the readers about the merit of the study.Author Response
Please see the attachment.

Reviewer 2 Report
The authors report on a monolithic gimbal micro-mirror which is fabricated and remotely tuned by a femtosecond laser. Both the advanced thermal management by variable ultrashort-pulse laser exposure and the high-precision contactless position control approach are very interesting for the community. Theoretical treatment based on von Mises stiffness matrix is performed. The paper is well written and can be recommended for a publication after minor improvements.
Comments:
(1) p. 1, lines 31/32: the phrase could be extended to “one of the most basic components in an adaptive optical system”
(2) p. 6, Fig. 3: The text within the pictures could better be recognized with white letters instead of black ones.
Typos:
- p. 4, line 156: “stiffness’s” has to be replaced by “stiffnesses”
- p. 4 line 170: “resonant frequencies” is often used in literature but, in the opinion of the reviewer, “resonance frequencies” sounds better matching
- p. 13, line 419: “better “have widely been used”
- List of references, [9]: should correctly be ”Opt. Mat. Express”
Round 2
Reviewer 1 Report
Dear Reviewer,
We are thankful for taking the time to review our paper. Below, you will find our answers to your points:
Point 1: 1 can be separated as (a) and (b).
Response 1: Since (a), (b), (c), (d) and (e) are used to refer to the different parts of the diagram on the right, we refer to the diagrams as “Left” and “Right” to avoid any confusion.
Q: They are exactly two individual graphs as subscripted. The authors can designate the two graphs as (i) and (ii), more precisely, (i) CAD….. (ii) Equivalent …. Also the axis of rotation can be denoted in the drawing too.
Point 2: 2 (c) is overlapped, and there is no color code for the
Response 2: We cannot see any overlap in the figure 2(c). However, if it feels too congested, we have rectified that a little. We believe the sentence in the review is incomplete as we do not understand which color code the reviewer is referring to.
Q: Fig. 2 (a) is overlapped with (c) and the spacing can be adjusted for better vision. The scale in Fig. 2(c) is not denoted in color code as in (a) or (b). The caption for (c) is capitalized which is different from (a) and (b).
Point 3: Also the stress distribution (a) does not show where the max static stress 12MPa is
Response 3: The old figure has been replaced now, clearly indicating where the maximum stress is.
Point 4: 1 Exposure Setup seems to be in section 2
Response 4: We do not understand what the reviewer means. As far as the setup is concerned, it is in Section 3 under the sub-section 3.1 titled as “Exposure Setup”
Q: In this section, it covers some contents about setup which are not shown in Device section. Section 3. is about Results, and now here comes sub-section 3.1 ….setup, isn’t it puzzling? (I suppose it’s associated with DEVICE. That’s whole lot of experimental apparatus).
Point 5: Line 209 repeated words
Response 5: The repetition has now been corrected.
Point 6: The whole paragraph should be rewritten for clarity
Response 6: We have re-written the paragraph, better explaining the exposure assembly and the use of the f-theta lens.
Point 7: The function of the f-theta lens is not clear. It may change the deflected angle to be measured.
Response 7: Since it is a challenging task to measure angular deviations as low as few micro-radians, we employ an f-theta lens. An advantage of using such a lens is that it has a flat focal plane, which means the beam is always focused in the same plane even when the input beam deviates off the optical axis of the lens, as is the case in this experiment. Additionally, an f-theta lens amplifies the final movement of the focal spot in the image plane as is suggested in equations 4, 5 and 7. The amplification factor is the focal length of the lens. One can then retrieve the actual angular displacement of the “device under test”, in our case, the rotation of our mirror, by dividing the output displacement on the detector by the focal length of the lens (Eq. 4,5,7).
The f-theta lens has no effect on the actual angle to be measured, other than amplifying it many times by a known factor (focal length). This factor when removed, gives the actual angular rotation.
All of this can be found in the new manuscript in lines 221-236 and 331-335.
Q: I can accept the argument, however, certain error will be induced for this part.
Point 7: The actuator is not explained in the manuscript.
Response 7: The explanation has now been added in lines 88-90. An already existing description about the actuator can be found in lines 112-116. With reference to Fig. 1 (right), further explanation can be found in lines 122-124
Point 8: I wonder if the femto second laser is used for fabrication purpose or as a driving force. It’s confusing. If for driving purpose, using ultrafast laser is an over-kill.
The femtosecond laser is used bothfor fabrication and as a source of actuation to re-align the mirror. However, we use different lasers for fabrication and exposure. As mentioned in Sec. 2.5, a short pulse laser with 270 fs pulses at 1030 nm wavelength is used for fabrication. To realign the mirror, we expose the actuator using two different fs pulses: one with a pulse width of 50 fs and another with a much longer pulse width of 300 fs (sections 3.3 and 3.4). This is done in order to access different modification regimes (referred in the manuscript as Regime I and Regime II).
To summarize, we use a fs laser for fabrication of the device. Later on, we use another fs laser, as a method of actuation (or “driving force”) with different parameters to re-align the mirror.
Femtosecond laser machining is a rather well-established technique for fabrication of 3-dimensional structures in transparent materials. There is not competing technologies to manufacture the device presented here in silica with the level of accuracy required. For the repositioning, it is necessary to induce a volume expansionthat becomes permanent after. (It is in essence a one-time actuation.) To modify a material in its volume and to cause this volume expansion without thermal effects, one needs to induce non-linear absorption phenomena, which can be obtained if high peak power is used, which is only easily accessible with femtosecond laser pulses.
Although static, multicomponent structures have been demonstrated mostly, we show that by combining the desired element with flexures, a very fine repositioning, with order of magnitude higher resolution, can be achieved in a contactlessmanner by using the laser as a tool. This paves the road towards miniaturized, monolithic fabrication of active components in miniaturized optical circuits, whereby the active component can be locally tuned/moved in order to achieve the ultimate goal. Such a tuning would hence be achieved without the need for positioners, non-invasively and with much higher precision.
Q: If the whole ablation process is under non-linear absorption mechanism the “heat effect” (including thermal expansion) will be minimized. For measuring the position, fs laser is an over-kill indeed unless the laser is tunable and used simultaneously for both fabrication and measurement. What we do in the field in using IR CCD or other precision camera on-line (Of course, it depends on what accuracy we acquired, less than 1 micrometer is quite common). Also, the authors use the terms: Heat affected and Laser affected zones, the difference between them should be explained to avoid confusing or just use one term is fine with detailed physical explanations. It's subtle though.
The manuscript involves lots of work and takes lengthy text to explain. The authors may need to proof read the whole manuscript and correct all the minor or confusing errors.
One last comment, the authors’ work is mostly on the mirror. Can any device without the similar mirror structure follow the same way?
